# Distinct Effects of High-Fat and High-Phosphate Diet on Glucose Metabolism and the Response to Voluntary Exercise in Male Mice

**DOI:** 10.3390/nu14061201

**Published:** 2022-03-12

**Authors:** Pablo Vidal, Lisa A. Baer, Elisa Félix-Soriano, Felix T. Yang, Daniel A. Branch, Kedryn K. Baskin, Kristin I. Stanford

**Affiliations:** Department of Physiology and Cell Biology, The Ohio State University Wexner Medical Center, Columbus, OH 43210, USA; pablo.vidalsouza@osumc.edu (P.V.); lisa.baer@osumc.edu (L.A.B.); elisa.felixsoriano@osumc.edu (E.F.-S.); felix.yang@osumc.edu (F.T.Y.); daniel.branch@osumc.edu (D.A.B.); kedryn.baskin@osumc.edu (K.K.B.)

**Keywords:** exercise, high-fat diet, high-phosphate diet, adipose tissue, metabolism

## Abstract

The prevalence of metabolic diseases is rapidly increasing and a principal contributor to this is diet, including increased consumption of energy-rich foods and foods with added phosphates. Exercise is an effective therapeutic approach to combat metabolic disease. While exercise is effective to combat the detrimental effects of a high-fat diet on metabolic health, the effects of exercise on a high-phosphate diet have not been thoroughly investigated. Here, we investigated the effects of a high-fat or high-phosphate diet in the presence or absence of voluntary exercise on metabolic function in male mice. To do this, mice were fed a low-fat, normal-phosphate diet (LFPD), a high-phosphate diet (HPD) or a high-fat diet (HFD) for 6 weeks and then subdivided into either sedentary or exercised (housed with running wheels) for an additional 8 weeks. An HFD severely impaired metabolic function in mice, increasing total fat mass and worsening whole-body glucose tolerance, while HPD did not induce any notable effects on glucose metabolism. Exercise reverted most of the detrimental metabolic adaptations induced by HFD, decreasing total fat mass and restoring whole-body glucose tolerance and insulin sensitivity. Interestingly, voluntary exercise had a similar effect on LFPD and HPD mice. These data suggest that a high-phosphate diet does not significantly impair glucose metabolism in sedentary or voluntary exercised conditions.

## 1. Introduction

Metabolic diseases including obesity and type 2 diabetes are rapidly increasing across the United States and worldwide [1]. Two extrinsic factors that contribute to the increased prevalence of metabolic disease are poor diet and lack of physical activity [2]. Over the last century, access to food has increased exponentially in developed countries, leading to significant changes in dietary patterns. Consumption of energy-rich foods, such as those with higher fat content, is one of the factors involved in the increasing rates of obesity [3]. The effects of a high-fat diet (HFD) on whole-body and white adipose tissue (WAT) function are well characterized [4,5,6,7,8]. HFD increases body weight, adipose tissue mass, and adipocyte size, eventually leading to impairments in whole-body glucose tolerance and insulin sensitivity [4,5,6,7].

Since the 1990s, consumption of phosphate additives has more than doubled [9], and almost 50% of all grocery products contain phosphate as a food additive, especially frozen food, dry food mixes and packaged meat [10]. Fast food and soda also contain high levels of phosphate additives [9,11]. A high-phosphate diet (HPD) detrimentally affects bone health [12], is a risk factor in cardiorenal syndrome [13], and is correlated with a higher all-cause mortality [14]. However, the effects of HPD on metabolic function, and specifically on white adipose tissue, remain largely understudied [15].

An important therapeutic approach to prevent the development of metabolic disease is exercise [16,17]. Chronic exercise results in systemic adaptations including improved glucose tolerance and increased energy expenditure, as well as tissue-specific adaptations to skeletal muscle, heart, and adipose tissue, among others [16,17,18]. Exercise induces a wide variety of adaptations to WAT; it increases glucose metabolism and lipolysis, decreases adipocyte size, and increases its endocrine activity [18,19,20,21,22,23]. Exercise prevents many of the detrimental adaptations induced by HFD in WAT [5,7,24,25,26,27,28,29]. In rodents, exercise prevents the diet-induced gain in fat mass [7,24,28,29] and reverses the detrimental effects of an HFD on glucose tolerance [7,27,28]. However, the role of exercise and a high phosphate diet (HPD) on WAT has not been thoroughly investigated.

Here, we investigated the effects of a high-fat or a high-phosphate diet on metabolic adaptations induced by chronic exercise on whole-body metabolism and subcutaneous WAT (scWAT). An HFD impaired metabolic function in mice, and chronic exercise partially rescued this phenotype by reducing body weight and adipose tissue mass and improving glucose and insulin tolerance. An HPD did not impair glucose metabolism in sedentary mice. Voluntary exercise significantly reduced adiposity in HPD-fed mice, but had a minimal effect on glucose metabolism. These results indicate that HFD, but not HPD, has a significant effect on whole-body metabolism, and that exercise reverts most of the detrimental adaptations induced by HFD.

## 2. Materials and Methods

### 2.1. Mice

Male, 20-week-old C57BL/6N mice from Charles Rivers or C57BL/6J Jackson Laboratory were used for the experiments [10]. Animals were maintained on a 12 h light/dark cycle at room temperature (22 °C). All procedures followed the Guide for the Care and Use of Laboratory Animals and following protocols approved by the IACUC at The Ohio State University. Animals were sacrificed using isoflurane followed by cervical dislocation prior to tissue collection.

### 2.2. High-Fat Diet and High-Phosphate Diet

Mice were fed a low-fat normal-phosphate diet (12.8% kcal from fat; 0.9% total Pi, TD160114, Envigo, Teklad Diets, Madison, WI, USA), high-fat diet (60% kcal from fat; 1.17% total Pi, Research Diets Inc., New Brunswick, NJ, USA), or high-phosphate diet (12.8% kcal from fat, 2.3% total Pi; TD08020, Envigo, Teklad Diets, Madison, WI, USA) for 16 weeks. Mice were fed *ad libitum* throughout the study. Mineral content for low-fat normal-phosphate and high-phosphate diet are the same (0.3% magnesium, 1.9% calcium, 1.8% potassium, and 0.9% sodium). Mineral content for high-fat diet (0.1% magnesium, 0.78% calcium, 0.26% potassium, and 0.13% sodium).

### 2.3. Wheel-Cage Training

After 6 weeks of diet intervention, mice were subdivided to either remain in static cages (sedentary) or given open access to a wheel cage (housed with running wheels where voluntary access to physical activity was available at all times) for a duration of 8 weeks.

### 2.4. Body Composition and Metabolic Testing

Body weight was measured every week using an OHAUS NV212 scale. An EchoMRI LLC was used to measure body fat and lean mass. Glucose tolerance tests were performed after an overnight 12-h fast with *ad libitum* drinking water. Exercised mice did not have access to the wheel overnight. Baseline blood glucose was measured by a tail-vein prick using a commercial glucose monitor (OneTouch Ultra2). Then, glucose was administered by intraperitoneal injection (2 g of glucose per kg of body weight) at 0 min, and blood from the same tail vein prick was used to measure blood glucose levels at 15-, 30-, 60-, and 120-min post injection. Insulin tolerance testing was performed following a 2 h fast in the afternoon with *ad libitum* drinking water. Exercised mice did not have access to the wheel during these 2 h. Baseline blood glucose was measured using a tail vein prick. Insulin was administered by intraperitoneal injection (1 unit per kg of body weight) at 0 min. Blood glucose was measured at 10, 15-, 30-, 45-, and 60-min post injection using the same tail vein prick. If at any time a mouse dropped below 40 mg/dL of blood glucose, they were removed from the test and given an intraperitoneal injection of 200 μL of 20% glucose to reverse the hypoglycemic state

### 2.5. Metabolic Chambers

Mice were housed in the Comprehensive Lab Animal Monitoring System (CLAMS, Oxymax Opto-M3; Columbus Instruments, Columbus, OH, USA) to measure volume of O_2_ consumption, CO_2_ production, respiratory exchange ratio (RER), and heat production. Data were collected over 48 h; for the first 24 h mice had *ad libitum* access to food and they were fasted the following 24 h [30].

### 2.6. Real Time Quantitative PCR

Tissue processing and quantitative PCR (qPCR) were performed as previously described [31]. Custom primers were used for genes of interest with the sequences shown in Appendix A. Gene expression data was normalized to the housekeeping gene *Gapdh*.

### 2.7. Histology Analysis

Subcutaneous white adipose tissue was removed from mice, immediately fixed in 10% formalin for 1 h, and then stained with hematoxylin-eosin. Images were taken under a GFP-B filter (487 nm) on a Nikon A1R (Nikon Instruments Inc., Tokyo, Japan). Three different cross-sectional areas per sample were analyzed for adipocyte area quantification using ImageJ software (National Center for Biotechnology Information, NCBI, Bethesda, MD, USA).

### 2.8. Statistical Analysis

Scientific software GraphPad Prism version 9 (GraphPad Software, San Diego, CA, USA) was used for data visualization and statistical analysis. The sample sizes and type of statistical test used of each experiment are described in the figure legends. Only mice that did not voluntarily run were excluded from the analyses (n = 3, LFPD group). Statistical significance was defined as *p* < 0.05.

## 3. Results

### 3.1. High-Fat, but Not High-Phosphate Diet, Has Detrimental Effects on Body Composition and Systemic Metabolism

To investigate the role of a high-fat or high-phosphate diet on systemic metabolism, twenty-week-old C57BL/6 male mice were fed a low fat and normal phosphate (LFPD; control), high-phosphate (HPD), or high-fat (HFD) diet for 6 weeks prior to undergoing metabolic analyses. HPD did not affect body weight, while HFD steadily increased body weight throughout the intervention (Figure 1A). After 6 weeks of diet intervention, there were no differences in lean mass among groups (Figure 1B), but HFD mice significantly increased fat mass compared to LFPD and HPD mice (Figure 1C). These results indicate that HFD increases fat mass while HPD does not affect adiposity.

Previous studies have shown that male mice fed an HPD for 12 weeks have decreased fat oxidation and elevated respiratory exchange ratio (RER) [15]. To determine the effect of HPD or HFD on overall metabolism after 6 weeks of diet, mice were placed in metabolic chambers. There were no differences in VO_2_ or VCO_2_ between HPD and control mice after 6 weeks on diet, while HFD mice had significantly lower VO_2_, and VCO_2_ (Figure 1D,E). Interestingly, HPD mice had increased RER in the fed state, while RER was decreased in HFD mice in both fed and fasted states (Figure 1F).

To compare the effects of an HPD and HFD on glucose metabolism, glucose and insulin tolerance tests were performed after 6 weeks of diet intervention. There was no effect of an HPD on glucose or insulin tolerance (Figure 2A–D). HFD-fed mice had significantly impaired glucose and insulin tolerance compared to both LFPD and HPD mice (Figure 2A–D). These data indicate that 6 weeks of HFD, but not HPD, impairs whole-body glucose metabolism and insulin tolerance.

### 3.2. Exercise Improves Body Composition and Metabolic Capacity in High-Fat Fed Mice

Exercise is an effective tool to treat obesity and the negative metabolic adaptations induced by HFD in mice [7,24,25,27,29]. It has been established that an HPD impairs exercise capacity during a graded maximal exercise test in mice [15], but the effect of chronic voluntary exercise on metabolic health in mice fed an HPD has not been thoroughly investigated. To determine the effect of chronic voluntary exercise on metabolic health in HFD and HPD mice, mice were maintained on their respective diets and subdivided into sedentary or exercise-trained for 8 weeks. Exercise-trained mice were given open access to a wheel cage. There was no difference in amount run among LFPD and HPD mice, but HFD mice ran significantly less over the 8-week exercise intervention (Figure 3A), similar to what has been previously observed [32,33].

Voluntary exercise reduced fat mass in all groups of exercised mice (LFPD, HPD, and HFD) and reduced body weight in LFPD and HFD mice (Figure 3B–D). There was no effect of LFPD, HPD, or HFD on bone mass normalized per lean mass (Figure 3E). Whole-body metabolic function was assessed with metabolic chambers; 8 weeks of exercise did not change VO_2_ or VCO_2_ in LFPD-fed or HPD mice. HFD decreased VO_2_ when compared to LFPD and HPD, and exercise restored it similar to the LFPD but only during the light cycle (Figure 3F–H). HPD increased RER, but exercise restored it to level of LFPD (Figure 3H). Together these data indicate that voluntary exercise has a minimal effect on whole-body metabolic function in HPD mice but reduces adiposity and body weight and improves whole-body metabolism in HFD mice.

To investigate the combined effects of exercise with LFPD, HPD, and HFD on glucose metabolism, glucose and insulin tolerance tests were performed after 14 weeks of diet and 8 weeks of voluntary exercise intervention. There was no effect of exercise in mice fed an LFPD or HPD diet, but glucose tolerance and insulin tolerance were significantly improved with exercise in the HFD-fed mice, restoring them to the level of LFPD-Sed (Figure 4A–H).

### 3.3. Exercise Reverses the Detrimental Effects of High-Fat Diet in Adipose Tissue

Exercise induces many positive adaptations to WAT [18,19,21,34]. Since exercise decreases total fat mass in all groups (Figure 3D), we investigated the specific adaptations of exercise to white adipose tissue. Exercise did not significantly change adipocyte size in LFPD and HPD mice. HFD induced a severe hypertrophy of the adipocytes, and exercise partially reversed this effect (Figure 5A–H). HPD did not significantly affect gene expression in adipose tissue under sedentary conditions, but with exercise, *Prdm16* expression is increased (Figure 6A). HFD severely decreases expression of many genes involved with metabolic processes, and exercise reverts them to normal levels (Figure 6B). There was no effect of HFD or HPD on inflammatory genes (Figure 6A,B).

## 4. Discussion

This study investigated the effects of exercise on LFPD, HPD and HFD in male mice. An HPD diet did not impair glucose metabolism in sedentary or voluntary exercised conditions. In contrast, an HFD significantly impaired metabolic function and voluntary exercise largely reverted most of the negative adaptations induced by obesity. These adaptations occur even though HFD mice run significantly less than LFPD and HPD mice.

Consistent with a previous study [15], we found that HPD increased RER after 6 to 14 weeks of diet. Interestingly, differences in RER were observed in the fed state but not the fasted state. Indeed, HPD significantly impairs fatty acid metabolism which can also contribute to higher RER [15]. We and others did not see any differences in gene expression of several markers in scWAT, indicating HPD does not affect adipose tissue to drive this increase in RER [15].

Higher serum phosphate levels are correlated to increased sedentary activity in humans, and 12 weeks of HPD decreases exercise capacity during a graded maximal exercise test in rodents [15]. We hypothesized that HPD would decrease the amount of distance mice would voluntarily run when housed with access to a wheel *ad libidum*, but there was no difference in running distance between LFPD and HPD after 8 weeks of voluntary wheel training. This may be a result of exercise intensity; voluntary wheel running is a low-intensity exercise and it is possible that high serum phosphate only impairs high-intensity exercise [35]. Since phosphate plays a key role in muscle contraction [36], it would not be surprising if high serum phosphate has a greater effect on muscle contractility in response to maximal intensity exercise.

Exercise and diet have significant effects on scWAT [18,19,20,21,27]. Our data shows that HFD induces many detrimental adaptations to scWAT, while this is not seen in HPD mice. Additionally, voluntary exercise reverses most of the detrimental effects of an HFD in scWAT, while it has a minimal effect on LFPD and HPD mice. It is important to note that both control and HPD mice were fed a diet low in fat (12.8% kcal from fat compared to the ~20%kcal from standard chow diets). Coincidentally, exercise-induced adaptations to metabolic health and white adipose tissue are blunted in a low-fat diet [34,37]. This could explain why there was a minimal effect of voluntary exercise in our control and HPD mice.

There are some limitations to this study. First, only male mice were used in the analysis, so it has not been established that female mice would have the same response to HPD in the presence or absence of exercise. In addition, we did not investigate the effects of HPD in combination with either standard fat (20% kcal/fat) or high-fat diet (60% kcal/fat) conditions. This will be important in the future, since it is well-known that exercise induces several adaptations in mice fed a standard chow diet [19,22,23].

To our knowledge, this is the first study to analyze the effect of HPD and HFD with chronic exercise and its metabolic adaptations. HPD does not impair voluntary exercise activity or affect whole-body glucose metabolism. The current data indicate that while a high-fat diet impaired metabolic health, a diet that is high-phosphate but low-fat did not affect glucose metabolism in either sedentary or exercised conditions.

## Figures and Tables

**Figure 1 nutrients-14-01201-f001:**
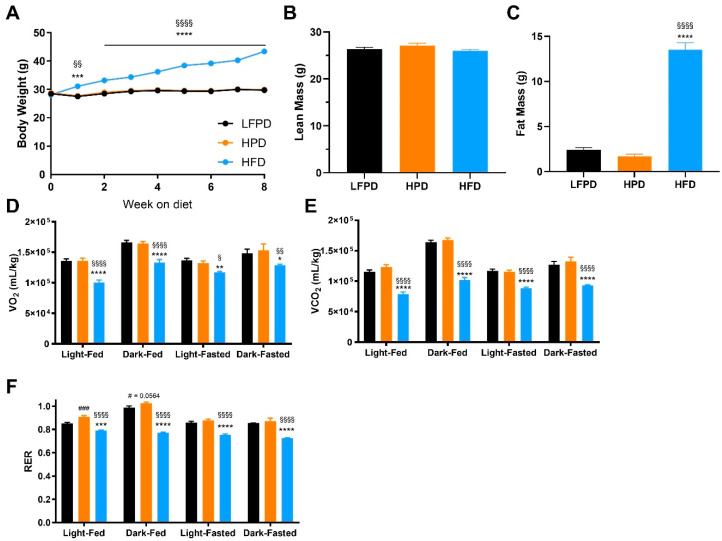
Six weeks of HFD impaired glucose metabolism. (**A**) Body weight, (**B**) Total lean mass, (**C**) Total fat mass, (**D**) Oxygen consumption (VO_2_), (**E**) Carbon dioxide production (VCO_2_), and (**F**) Respiratory Exchange Ratio after 6 weeks of diet intervention. N = 14–30 per group for (**A**–**C**), n = 7–10 per group for (**D**–**F**). One or two-way ANOVA were used with Tukey’s multiple comparison test (* *p* < 0.05, ** *p* < 0.01, *** *p* < 0.001, **** *p* < 0.0001 LFPD vs. HFD, ^§^ *p* < 0.05, ^§§^ *p* < 0.01, ^§§§§^ *p* < 0.0001 HPD vs. HFD, ^#^ *p* < 0.05, ^###^ *p* < 0.001 LFPD vs. HPD). Data are expressed as the mean ± SEM.

**Figure 2 nutrients-14-01201-f002:**
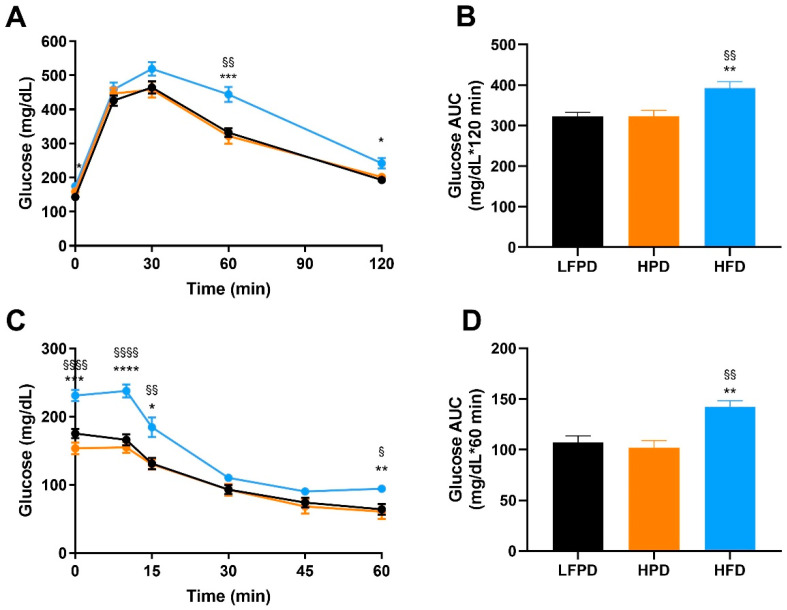
Six weeks of HFD impaired whole-body glucose tolerance and insulin sensitivity. (**A**) Glucose tolerance test exclusion curve, (**B**) Glucose tolerance area under the curve (AUC), (**C**) Insulin tolerance test exclusion curve, and (**D**) Insulin tolerance AUC. N = 14–30 per group. One or two-way ANOVA were used with Tukey’s multiple comparison test (* *p* < 0.05, ** *p* < 0.01, *** *p* < 0.001, **** *p* < 0.0001 LFPD vs. HFD, ^§^ *p* < 0.05, ^§§^ *p* < 0.01, ^§§§§^ *p* < 0.0001 HPD vs. HFD). Data are expressed as the mean ± SEM.

**Figure 3 nutrients-14-01201-f003:**
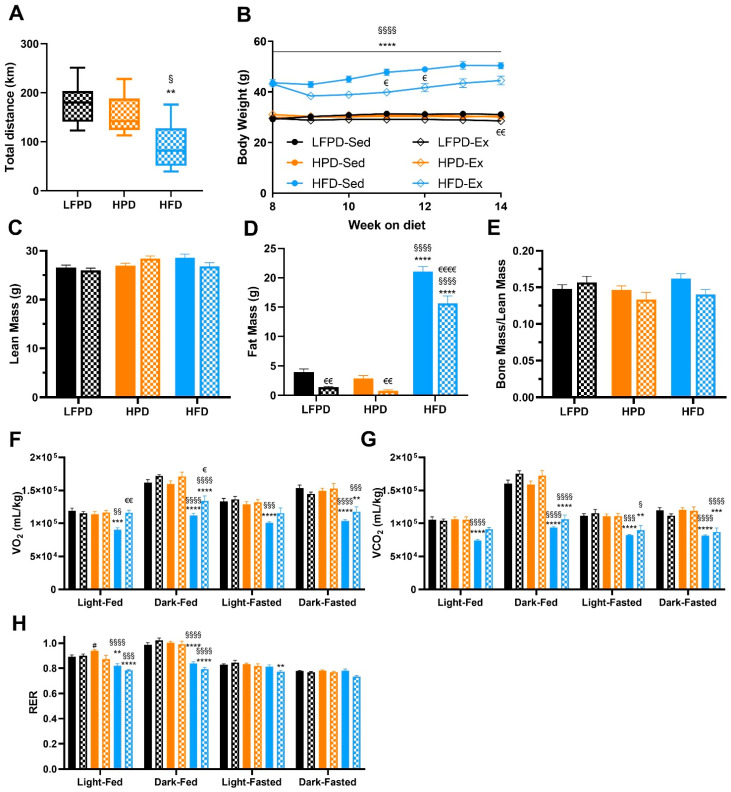
Eight weeks of exercise significantly improves body composition and reverses the detrimental effects of an HFD in oxygen consumption. (**A**) Total distance run, (**B**) Body weight, (**C**) Total lean mass, (**D**) Total fat mass, (**E**) Bone mass normalized to lean mass, (**F**) Oxygen consumption (VO_2_), (**G**) Carbon dioxide production (VCO_2_), and (**H**) Respiratory Exchange Ratio. N = 7–15 per group). T-test or one or two-way ANOVA were used with Tukey’s multiple comparison test (** *p* < 0.01, *** *p* < 0.001, **** *p* < 0.0001 LFPD vs. HFD, ^§^ *p* < 0.05, ^§§^ *p* < 0.01, ^§§§^
*p* < 0.001 ^§§§§^ *p* < 0.0001 HPD vs. HFD, ^#^ *p* < 0.05 LFPD vs. HPD, ^€^ *p* < 0.05, ^€€^ *p* < 0.01, ^€€€€^ *p* < 0.0001 Sedentary vs. Exercise). Data are expressed as the mean ± SEM. Box and whisker plot is shown as min to max with median.

**Figure 4 nutrients-14-01201-f004:**
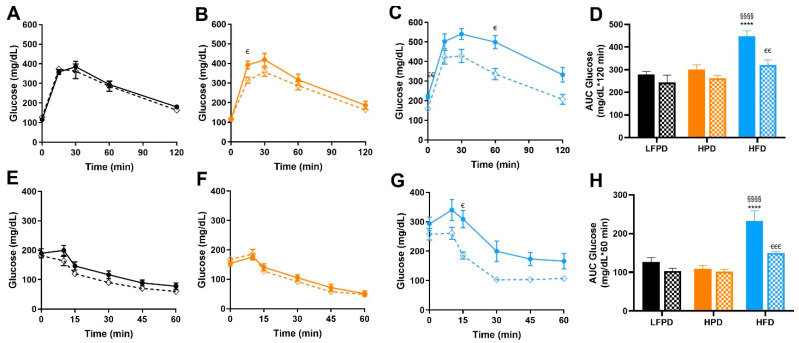
Exercise improves whole-body glucose tolerance and insulin sensitivity in HFD mice. (**A**) Glucose tolerance test exclusion curve for LFPD mice, (**B**) Glucose tolerance test exclusion curve for HPD mice, (**C**) Glucose tolerance test exclusion curve for HFD mice, (**D**) Glucose tolerance area under the curve for all groups of mice (**E**) Insulin tolerance test exclusion curve for LFPD mice, (**F**) Insulin tolerance test exclusion curve for HPD mice, (**G**) Insulin tolerance test exclusion curve for HFD mice, and (**H**) Insulin tolerance area under the curve for all groups of mice after 6 weeks of exercise. N = 7–15 per group. One or two-way ANOVA were used with Tukey’s multiple comparison test (**** *p* < 0.0001 LFPD vs. HFD, ^§§§§^ *p* < 0.0001 HPD vs. HFD, ^€^ *p* < 0.05, ^€€^ *p* < 0.01, ^€€€^ *p* < 0.001 Sedentary vs. Exercise). Data are expressed as the mean ± SEM.

**Figure 5 nutrients-14-01201-f005:**
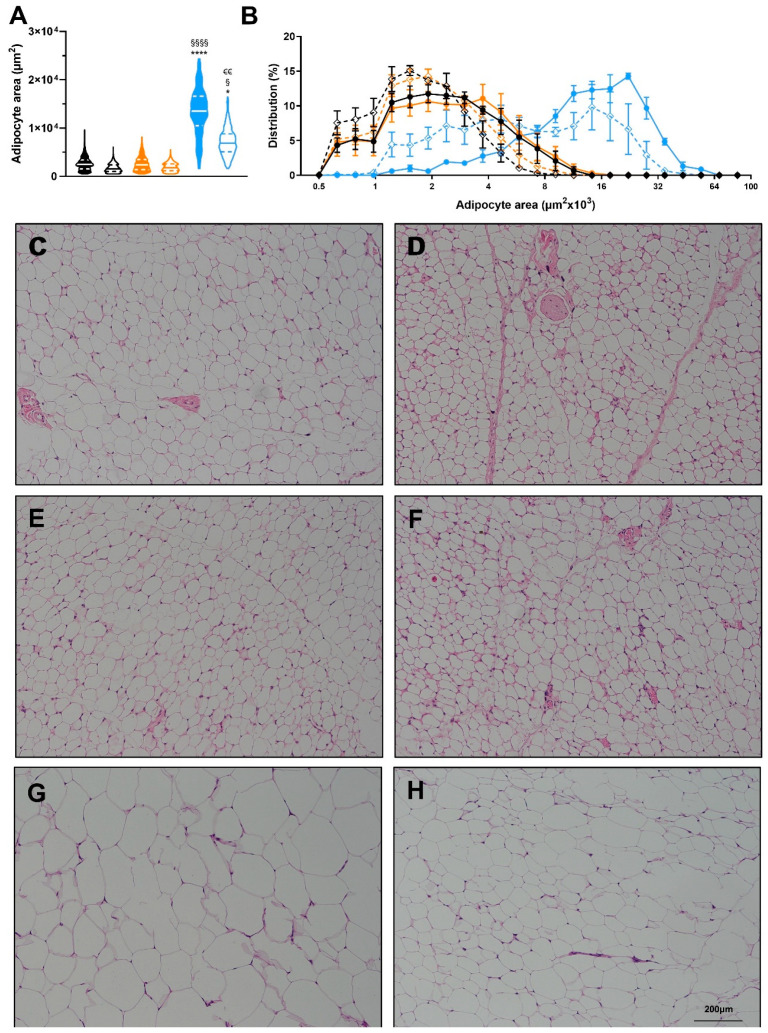
Effects of diet and exercise on adipocyte size. (**A**) Average adipocyte area of scWAT, (**B**) Frequency distribution of adipocyte area of scWAT. Representative histology images for (**C**) LFPD-Sed, (**D**) LFPD-Ex, (**E**) HPD-Sed, (**F**) HPD-Ex, (**G**) HFD-Sed and (**H**) HFD-Ex. N = 3–4 per group). Nested one-way ANOVA was used with Tukey’s multiple comparison test in (**A**). One or two-way ANOVA were used with Tukey’s multiple comparison test in (**B**). (* *p* < 0.05, **** *p* < 0.0001 LFPD vs. HFD, ^§^ *p* < 0.05, ^§§§§^ *p* < 0.0001 HPD vs. HFD, ^€€^ *p* < 0.01 Sedentary vs. Exercise). Data are expressed as the mean ± SEM. Violin plot is shown as min to max with median and first and third quartiles.

**Figure 6 nutrients-14-01201-f006:**
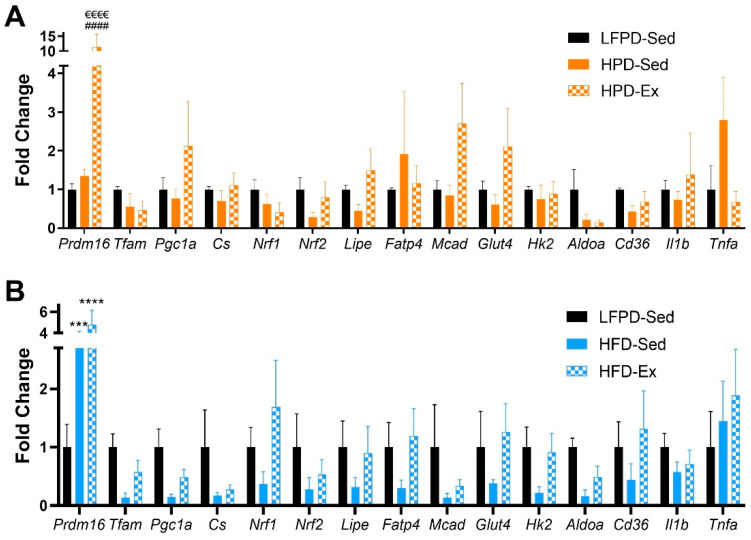
Exercise affects expression of metabolic and inflammatory genes in scWAT. (**A**,**B**) Gene expression data of scWAT (n = 3–7 per group). One-way ANOVA was used with Tukey’s multiple comparison test. (*** *p* < 0.001, **** *p* < 0.0001 LFPD vs. HFD, HPD vs. HFD, ^####^ *p* < 0.0001 LFPD vs. HPD, ^€€€€^ *p* < 0.0001 Sedentary vs. Exercise). Data are expressed as the mean ± SEM.

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
