# Peer review of "Distinct Effects of High-Fat and High-Phosphate Diet on Glucose Metabolism and the Response to Voluntary Exercise in Male Mice"

_nutrients, 2022, doi:10.3390/nu14061201_

Round 1
Reviewer 1 Report
The authors conducted a study entitled, "Distinct effects of high-fat and high-phosphate diet on glucose 2 metabolism and the response to voluntary exercise in male 3 mice." The authors state that the effects of exercise on a high-phosphate diet have not been thoroughly investigated. Therefore, they investigated the effects of a high-fat or high-phosphate diet in the presence or absence of voluntary exercise on metabolic function in male mice. The manuscript was very well written and addresses a needed question with regards to high phosphate diets. However, I have two major criticisms of this study:
- No female mice were studied, and
- There was no combination treatment ( HFD plus HPD).
Therefore, these limitations should be addressed in the discussion.
Author Response
Reviewer 1:
The authors conducted a study entitled, "Distinct effects of high-fat and high-phosphate diet on glucose 2 metabolism and the response to voluntary exercise in male 3 mice." The authors state that the effects of exercise on a high-phosphate diet have not been thoroughly investigated. Therefore, they investigated the effects of a high-fat or high-phosphate diet in the presence or absence of voluntary exercise on metabolic function in male mice. The manuscript was very well written and addresses a needed question with regards to high phosphate diets. However, I have two major criticisms of this study:
- No female mice were studied, and
- There was no combination treatment (HFD plus HPD).
Therefore, these limitations should be addressed in the discussion.
ïƒ Thank you for these comments. We have added a comment in the discussion to state that there are some limitations to the study in that only male C57BL/6 mice were used, and there was no combination treatment of HFD plus HPD.

Reviewer 2 Report
The study by Vidal P et al., entitled – “Distinct effects of high-fat and high-phosphate diet on glucose 2 metabolism and the response to voluntary exercise in male 3 mice” is an interesting study but needs further validation. The suggested experiments will definitely strengthen the data and make it suitable for publication.
The comments are attached below.
1 . In figure -2 the authors have measured whole body glucose and insulin tolerance. Just from these measurements it won’t be clear whether the glucose disposal is happening via skeletal muscle and liver glucose uptake ? or is there any altered function with respect to hepatic gluconeogenesis? It is ideal to measure under the following time points the plasma insulin and glucagon levels and also skeletal muscle and liver insulin signaling along with GLUT4 translocation? We need to have a clear sense of how this disposal is happening.
2 . In Figure 4, following exercise stimulation it is ideal to measure total and phosphorylated AMPK levels and include it.
3 . In Figure 5, following exercise stimulation it is important to measure GLUT4 translocation to plasma membrane either by IHC/IF techniques or by Western blot using whole tissue and membrane fractions. This will really strengthen the data.
4 . The impact of high phosphate diet is not fully known, so it will be ideal to measure plasma AST and ALT levels as a mark for hepatotoxic effects following 6 weeks administration of this diet along with creatinine levels as a renal marker. If possible, it is ideal to measure bone density if the samples are available.
5 . To understand the impact of this diet on metabolic health and cardiovascular health, it is ideal to measure systemic inflammatory markers in plasma (C-reactive protein, IL-6) and also tissue specific expression of IL-6, TNF alpha and IL-1 Beta (heart and liver). Also, it is important to check whether high phosphate diet is triggering cell death especially apoptosis, autophagy or necrosis, so it is important to measure markers of apoptosis, autophagy and necrosis in these tissues. Please include that data as a supplemental figure.
Author Response
Reviewer 2:
The study by Vidal P et al., entitled – “Distinct effects of high-fat and high-phosphate diet on glucose 2 metabolism and the response to voluntary exercise in male 3 mice” is an interesting study but needs further validation. The suggested experiments will definitely strengthen the data and make it suitable for publication.
The comments are attached below.
1 . In figure -2 the authors have measured whole body glucose and insulin tolerance. Just from these measurements it won’t be clear whether the glucose disposal is happening via skeletal muscle and liver glucose uptake? or is there any altered function with respect to hepatic gluconeogenesis? It is ideal to measure under the following time points the plasma insulin and glucagon levels and also skeletal muscle and liver insulin signaling along with GLUT4 translocation? We need to have a clear sense of how this disposal is happening.
ïƒ Glucose and insulin tolerance tests were performed to determine if there was a difference among mice fed a LFPD, HPD, or HFD. There was no difference in glucose tolerance among LFPD or HPD mice. An HFD impaired glucose and insulin tolerance, as has been previously established (Kim et al, 2007; Gollisch et a, 2009, Martinez-Huenchullan et al 2019). Thus, it is not clear what measuring difference in muscle or liver tissue glucose disposal would add, as there is no difference indicated among LFPD or HPD mice and it has been previously indicated that glucose disposal is impaired in several tissues in the presence of an HFD (Atkinson et al, 2013; Gurley et al, 2016; Martinez-Huenchullan et al, 2019, Reilly et al, 2022).
2 . In Figure 4, following exercise stimulation it is ideal to measure total and phosphorylated AMPK levels and include it.
ïƒ Measuring AMPK and pAMPK are important indicators of exercise stimulation. As this was a chronic (6 wks) exercise training program, we measured exercise distance and detected multiple metabolic adaptations to exercise (improved glucose tolerance, reduced adiposity, reduced RER) (Martinez-Huenchullan et al, 2019; Raun et al, 2021; Nigro et al, 2021) indicating that the mice were exercise-trained.
3 . In Figure 5, following exercise stimulation it is important to measure GLUT4 translocation to plasma membrane either by IHC/IF techniques or by Western blot using whole tissue and membrane fractions. This will really strengthen the data.
ïƒ We agree with the reviewer that GLUT4 is one of many indicators of exercise-training adaptations. In this manuscript, we have demonstrated that the mice were exercise-trained by presenting distance run (km) and multiple metabolic adaptations to exercise (improved glucose tolerance, reduced adiposity, reduced RER) (Martinez-Huenchullan et al, 2019; Raun et al, 2021; Nigro et al, 2021), all indicating that the mice were exercise-trained.
4 . The impact of high phosphate diet is not fully known, so it will be ideal to measure plasma AST and ALT levels as a mark for hepatotoxic effects following 6 weeks administration of this diet along with creatinine levels as a renal marker. If possible, it is ideal to measure bone density if the samples are available.
ïƒ Previous work investigating the effects of a high-phosphate diet showed that it did not affect creatinine or creatinine clearance (Peri-Okonny et al, 2019). Recent studies showed that mice fed an HPD for 58 weeks did not have any detrimental effects on hepatic (Ugrica et al, 2022) or renal (Ugrica et al, 2021) function, although it did result in reduced bone density (Ugrica et al, 2021).
We have now included data measuring bone mass among mice and found that there was no difference in bone mass normalized per lean mass in HPD mice compared to all other groups (Fig. 3E).
5 . To understand the impact of this diet on metabolic health and cardiovascular health, it is ideal to measure systemic inflammatory markers in plasma (C-reactive protein, IL-6) and also tissue specific expression of IL-6, TNF alpha and IL-1 Beta (heart and liver). Also, it is important to check whether high phosphate diet is triggering cell death especially apoptosis, autophagy or necrosis, so it is important to measure markers of apoptosis, autophagy and necrosis in these tissues. Please include that data as a supplemental figure.
ïƒ The overall findings from this manuscript are that an HPD did not affect glucose metabolism. It has been previously shown that an HPD of a similar duration impairs fatty acid metabolism but did not affect body weight or cardiac contractile function (Peri-Okonny et al, 2019). Measurements of cardiac health and function are important, but beyond the scope of this manuscript.
To address the effects of an HPD on inflammatory markers in subcutaneous white adipose tissue, we have now included qPCR data measuring TNFa and IL1B and show that these genes are not altered by an HPD or HFD in sedentary or exercise-trained mice. These data are now included in Fig. 6A and B.
